# The Combination of GS-441524 (Remdesivir) and Ribavirin Results in a Potent Antiviral Effect Against Human Parainfluenza Virus 3 Infection in Human Airway Epithelial Cell Cultures and in a Mouse Infection Model

**DOI:** 10.3390/v17020172

**Published:** 2025-01-26

**Authors:** Yuxia Lin, Birgit Weynand, Xin Zhang, Manon Laporte, Dirk Jochmans, Johan Neyts

**Affiliations:** 1KU Leuven Department of Microbiology, Immunology and Transplantation, Rega Institute for Medical Research, Virology, Antiviral Drug & Vaccine Research Group, B-3000 Leuven, Belgium; yuxia.lin@kuleuven.be (Y.L.); zhang.xin007@wuxibiologics.com (X.Z.); manon.laporte@kuleuven.be (M.L.); 2KU Leuven Department of Imaging and Pathology, Division of Translational Cell and Tissue Research, B-3000 Leuven, Belgium; birgit.weynand@kuleuven.be; 3VirusBank Platform, KU Leuven, B-3000 Leuven, Belgium

**Keywords:** parainfluenza viruses, human respirovirus 3, remdesivir, obeldesivir, ribavirin, AG129 mice, human organoids

## Abstract

Human parainfluenza virus type 3 (HPIV-3) can cause severe respiratory diseases, particularly in young children, the elderly and immunocompromised. There are no approved antiviral drugs against this virus. We report that the combination of ribavirin with either remdesivir or its parent nucleoside GS-441524 results in a pronounced antiviral effect against HPIV-3 in LLC-MK2 cells and in human airway epithelial cells grown at the air–liquid interface. In AG129 mice intranasally inoculated with HPIV-3, the combined treatment with ribavirin and GS-441524 decreased infectious viral lung titers by >2.5 log_10_ to undetectable levels in 4 out of 11 mice and by 1.6 log_10_ in the remaining 7 mice as compared with the vehicle. The lungs of all mice that received the combined treatment appeared histologically normal or virtually normal, whereas 8 of 11 vehicle-treated mice presented with bronchopneumonia. By contrast, ribavirin alone did not result in a reduction in infectious viral lung titers; GS-441524 alone reduced infectious viral lung titers by 1.2 log_10_. Moreover, several mice in the single-treatment groups exhibited severe lung pathology. These findings may warrant exploring this combination in patients with severe HPIV-3 infections and possibly also against infections with other viruses that are susceptible in vitro to these two drugs.

## 1. Introduction

Human parainfluenza viruses (HPIVs), single-stranded enveloped RNA viruses of the family *Paramyxoviridae*, are responsible for common acute upper and lower respiratory infections across all age groups, with the highest infection rate in young children. HPIV infections can cause severe diseases such as croup, bronchiolitis and pneumonia [1,2]. Of the four HPIV serotypes, HPIV-3 is the most virulent, mainly causing lower respiratory tract illnesses such as bronchiolitis and pneumonia in children under 5 years old and the elderly. Infection with this virus may even be fatal in immunocompromised patients, particularly in hematopoietic stem cell transplant recipients, lung transplant recipients and children with severe combined immunodeficiency disease syndrome [3,4,5,6,7]. There are no approved antiviral drugs for the treatment of HPIV-3 infection. We therefore explored the repurposing of antiviral drugs, either alone or in combination, against infections with this virus. Several antiviral drugs have been investigated for repurposing as treatments for HPIV-3 infection, including the RNA-dependent RNA polymerase (RdRp) inhibitors remdesivir, ribavirin, and favipiravir, as well as the neuraminidase inhibitor zanamivir [8,9]. The latter drug results in a synergistic in vitro antiviral effect when combined with the entry inhibitor suramin [9].

Ribavirin (1-β-d-ribofuranosyl-1,2,4-triazole-3-carboxamide), a guanosine analog, has broad-spectrum antiviral activity against a variety of DNA and RNA viruses [10,11,12]. Ribavirin has been used to treat severe infections with Lassa fever virus, hepatitis C virus (HCV), respiratory syncytial virus (RSV) and parainfluenza viruses including HPIV-3 [13,14]. A few cases have been reported where ribavirin demonstrated a therapeutic effect in clearing HPIV-3 infections [15,16,17]. In two small uncontrolled series involving RSV/HPIV-3 infections in recipients of allogeneic stem cell transplantation, ribavirin therapy enabled most patients to recover from the infection, with deaths occurring only in those with severe respiratory dysfunction who required mechanical ventilation [6,18]. A recent comparative analysis revealed that ribavirin therapy resulted in a significant protective effect against progression from upper to lower respiratory tract HPIV-3-induced disease as compared to untreated cohorts [19].

Remdesivir (Veklury, Gilead Sciences), a monophosphoramidate nucleotide analog prodrug originally developed to treat HCV infections [20], has been approved for use in several countries to treat patients with COVID-19 [21]. GS-441524 is the parent nucleoside of remdesivir. Intracellularly, GS-441524 is converted to its 5ʹ-triphosphate metabolite, which, as an adenosine 5’-triphosphate mimic, inhibits the viral RdRp [22]. Remdesivir and GS-441524 inhibit the in vitro replication of a wide range of RNA viruses such as, but not limited to RSV, parainfluenza viruses, henipaviruses and coronaviruses [23]. Antiviral activity has also been demonstrated for several of these viruses in animal infection models [24,25,26,27].

The combined use of antiviral drugs has been particularly successful in treating infections with the human immunodeficiency virus (HIV) and HCV [13,28,29,30,31]. Here, we investigated the antiviral efficacy of combined treatment of ribavirin with either remdesivir or GS-441524 against HPIV-3 infection in LLC-MK2 cells, nasal human airway epithelial cells and AG129 mice.

## 2. Materials and Methods

### 2.1. Cells and Viruses

LLC-MK2 cells (rhesus monkey kidney, ATCC, CCL-7) were cultured in DMEM/F-12 medium supplemented with 10% fetal bovine serum and 1.5% HEPES at 37 °C in a 5% CO_2_ incubator. Endpoint titration assays were performed using a medium containing 2% fetal bovine serum instead of 10%.

HPIV-3-GFP was purchased from ViraTree. Clinical isolate of HPIV-3, “isolate 10-012854”, was kindly provided by Professor Frank Coenjaerts. The virus was obtained from a nasopharyngeal wash collected from a patient in March 2010 at the University Medical Center in Utrecht, the Netherlands [27]. HPIV-3-GFP and HPIV-3-isolate 10-012854 stocks used in this study were propagated in LLC-MK2 cells, which have titers of 1.3 × 10^8^ TCID_50_/mL and 3.8 × 10^7^ TCID_50_/mL, respectively.

### 2.2. Animals

AG129 mice (129/Sv mice with knockouts in both IFNα/β and IFNγ receptor genes) were bred in-house, with the breeding couple purchased from Marshall BioResources (Strasbourg, France). Female and male AG129 mice aged 8–12 weeks were used throughout the study. The specific pathogen-free status of the mice was regularly checked at the KU Leuven animal facility of the Rega Institute. Animals were housed in individually ventilated isolator cages (IsoCage N Biocontainment System, Tecniplast, West Chester, PA, USA) at a temperature of 21 °C and humidity of 55%, and subjected to 12:12 dark/light cycles. They had access to food and water ad libitum, and their cages were enriched with cotton and cardboard play tunnels for mice. Housing conditions and experimental procedures were approved by the Ethical Committee Dierproeven of KU Leuven (License P099/2022), following institutional guidelines approved by the Federation of European Laboratory Animal Science Associations. Mice were anesthetized by inhalation with isoflurane before infection and killed by intraperitoneally administering 100 µL of Dolethal before sample collection. Mice were monitored daily for any signs of disease.

### 2.3. Antiviral Molecules

GS-441524 was purchased from Excenen Pharmatech Co., Ltd. (Guangzhou, China). Ribavirin used in LLC-MK2 cell and human airway epithelial cell cultures was purchased from MedChemExpress (Monmouth Junction, NJ, USA). Ribavirin used for the mouse studies was purchased from BIOSYNTH (Staad, Switzerland). In LLC-MK2 cell experiments, a 100 mM stock solution of each molecule was prepared in analytical grade dimethyl sulfoxide (DMSO, Sigma, Burlington, MA, USA). In human airway epithelial cell culture experiments, a 10 mM stock solution of each molecule was prepared in DMSO. For mouse studies, GS-441524 was formulated as 0.75 mg/mL, 3.0 mg/mL and 7.5 mg/mL solutions in 30% PEG_400_ (Sigma) in PBS containing 1% DMSO. Ribavirin was formulated as 7.5 mg/mL and 15 mg/mL solutions in PBS.

### 2.4. Antiviral Assays In Vitro

LLC-MK2 cells were seeded in 96-well plates at a density of 1 × 10^4^ cells/well one day before infection. On the following day, cells were infected with HPIV-3-GFP at a final multiplicity of infection (MOI) of 0.1 in the presence of molecules. The molecules were added to cells in a matrix of six concentrations of ribavirin and seven concentrations of remdesivir or GS-441524. The intensity of GFP signals at day 3 post-infection (p.i.) was used as a readout to reflect viral replication levels. The percentage of inhibition was calculated by subtracting the background of the intensity of GFP signals in uninfected untreated control wells and normalizing it to the infected untreated control wells.

### 2.5. Quantitative Drug Combination Analysis

The Bliss scores of combinations were determined by the Bliss independence model to quantitatively assess molecule interaction patterns. The Bliss expected inhibition for a combined response was calculated by E = (A + B) − (A × B), where A and B are the fractional inhibition of HPIV-3 replications of molecule A and B at a given concentration. The Bliss score was calculated as the difference between the observed and expected inhibition of HPIV-3 replication under the same condition. Bliss scores between −10 and 10 indicate additive; Bliss scores >10 indicate synergy, where the activity is greater than additive; and Bliss scores < −10 indicate antagonism, where the activity is less than additive. The interaction between the molecules was also assessed using SynergyFinder v3.0 [32].

### 2.6. Viral Infection of Nasal Human Airway Epithelial Cell Culture

Nasal human airway epithelial cell (HnAEC) cultures (MucilAir, Cat. No. EP02MP) from a pool of donors were obtained from Epithelix (Plan-les-Ouates, Switzerland) in an air–liquid interphase setting and cultured in MucilAir medium (Epithelix, Cat. No. EP04MM) at 37 °C in a 5% CO_2_ incubator. A 100 µL inoculum of HPIV-3-GFP at a dose of 2.5 × 10^5^ TCID_50_/insert was added to the apical side. The molecules were added to the medium at the basolateral side. In the prophylactic setting, the apical site of the cultures was washed once with a pre-warmed MucilAir medium, and the inserts were transferred to a fresh medium with or without molecules. After 2 h of pre-incubation, the cultures were infected with HPIV-3-GFP for 2 h, after which the inoculum was removed. In the therapeutic setting, the cultures were washed once with a pre-warmed MucilAir medium and infected with HPIV-3-GFP for 2 h, followed by removal of the inoculum. Starting from day 4 p.i., the cultures were treated with or without molecules. At the indicated time points, the apical sides were washed with 250 µL pre-warmed medium and the medium at the basolateral side was refreshed with or without molecules. All apical washes were collected and stored at −80 °C for subsequent quantification of viral RNA levels by RT-qPCR or infectious virus titers by endpoint titration assays.

### 2.7. Quantitative Reverse Transcription-PCR (RT-qPCR)

RT-qPCR was performed on a LIGHTcycler96 platform (Roche Diagnostics, Diegem, Belgium) using the iTaq universal probes one-step kit (Bio-Rad, Cat. No. 1725141, Temse, Belgium). The primers targeting the *HN* gene of HPIV-3 encoding hemagglutinin-neuraminidase were forward primer 5′-*GGAGGTCTTGAACATCCAAT*-3′ and reverse primer 5′-*AGCCTTTGTCAACAACAATG*-3′. The probe was 5′-56-FAM/*AGAGACTGT*/ZEN/*AATCAAGCGTC*/3IABkFQ-3′. Equivalent standard serial dilutions of the HPIV-3 virus stock were used to quantify viral RNA levels per milliliter of apical wash in HnAEC cultures.

### 2.8. Cytotoxicity Measurement

Molecule/virus-induced cytotoxicity in the HnAEC culture was determined by detecting lactate dehydrogenase (LDH) released into the basolateral medium using CyQUANT LDH Cytotoxicity Assay kit (Invitrogen, Cat. No. C20300, Waltham, MA, USA). Briefly, 50 μL of basolateral medium of each sample was transferred into a 96-well plate and mixed with 50 μL of reaction buffer. Samples were incubated at room temperature for 30 min, protected from light, followed by an addition of 50 μL of stop solution to each well. Medium from HnAEC cultures that were lysed by Triton X-100 10% overnight was used as maximal LDH activity control. In parallel, fresh MucilAir medium served as background LDH activity control and an LDH positive control was included in each measurement. The absorbance at 490 nm and 680 nm were measured. The 680-nm absorbance values were subtracted from the 490-nm absorbance values to calculate the LDH activity. The percentage of cytotoxicity was determined using the formula: % cytotoxicity = (Sample LDH activity − Medium LDH activity)/(Maximal LDH activity − medium LDH activity) × 100.

### 2.9. Endpoint Virus Titration Assay

The endpoint virus titration assay was performed on LLC-MK2 cells in 96-well plates. GFP signals of HPIV-3-GFP were read out by SPARK (Tecan, Männedorf, Switzerland). The cytopathic effect induced by HPIV-3-isolate 10-012854 infection was scored under the microscope. Infectious virus titers were calculated using the Spearman Kärber method and expressed as the TCID_50_ per insert of HnAEC cultures and per milligram of lung sample.

### 2.10. Mouse Studies

The AG129 mouse infection model of HPIV-3 has been established in our laboratory [27]. AG129 mice were administered either the vehicle or drugs twice daily (BID) via oral gavage from day −1 to the day before the endpoint. At day 0, mice were anesthetized with isoflurane and intranasally inoculated with 40 µL of medium containing 1.5 × 10^6^ TCID_50_ HPIV-3-isolate 10-012854. Mice were monitored daily for weight loss and any clinical signs. At day 3 p.i., mice were euthanized to collect lung samples for quantification of infectious virus titers. At day 6 p.i., mice were euthanized to collect lung samples, and a portion of the lung was fixed in 4% formaldehyde for subsequent histological analysis. Lung samples were homogenized using bead disruption (Precellys, Bertin Corp., Rockville, MD, USA) in 400 µL medium, followed by centrifugation (10,000× *g*, 5 min) to pellet cell debris. The supernatant was then processed for endpoint virus titration assay.

### 2.11. Histology

Lung samples were fixed overnight in 4% formaldehyde and embedded in paraffin. Tissue sections (5 μm) were stained with H&E and scored blindly for lung damage by an expert pathologist as previously described [27]. Briefly, the five parameters, congestion, intra-alveolar hemorrhages, intra-alveolar edema, apoptotic bodies in the bronchus wall, and perivascular edema, were scored 0 or 1 for absent or present. The three parameters perivascular inflammation, peribronchial inflammation, and vasculitis were scored on a scale of 0 to 3, depending on the number of inflammatory cells. Bronchopneumonia was scored on a scale of 0–3 depending on the extent of the pathology: 1 is ≤40%, 2 is between 41% and 60%, and 3 is ≥61% of the affected lung area [33]. A cumulative score was calculated by summing the scores of each parameter. The higher the number, the worse the pathology. The max score is 17.

### 2.12. Statistical Analysis

All statistical analysis were performed using GraphPad Prism 10 (GraphPad Software, Inc., San Diego, CA, USA). Statistical significance was determined using an ordinary one-way ANOVA with Dunnett’s multiple comparisons test (human airway epithelial cell culture data), the non-parametric Kruskal–Wallis test with Dunn’s multiple comparisons test for multiple comparisons (mouse study data) or the Mann–Whitney *U* test for pairwise comparisons (mouse study data). *p*-values of <0.05 were considered significant. Nonsignificant, “ns”, indicates *p* > 0.05. Asterisks indicate a statistical significance level of * *p* < 0.05, ** *p* < 0.01, *** *p* < 0.001, **** *p* < 0.0001.

## 3. Results

### 3.1. Combinations of Ribavirin and Remdesivir or GS-441524 Result in Synergistic Antiviral Efficacy Against HPIV-3 Replication in LLC-MK2 Cells

We first investigated the antiviral activity of single treatments of either ribavirin, remdesivir and GS-441524 against HPIV-3, using a reporter virus that expresses enhanced green fluorescent protein (GFP), referred to as HPIV-3-GFP [34]. Ribavirin, remdesivir and GS-441524 inhibit HPIV-3 replication in LLC-MK2 cells in a dose-dependent manner, with EC_50_ of respectively 29 µM, 0.21 µM and 1.4 µM (Appendix A). Next, the antiviral activity and the toxicity effects of combinations of ribavirin with either remdesivir or GS-441524 were assessed in the same checkerboard format (Figure 1A,D). Both of the combinations presented additive effects between the two molecules (Appendix A). Bliss scores reveal that both combinations resulted in significant synergistic antiviral efficacy against HPIV-3 across broad concentration ranges of either molecule (Figure 1B,E). For instance, the combination of 12.5 µM ribavirin with 0.13 µM remdesivir resulted in 100% inhibition of HPIV-3 replication, whereas the Bliss model predicts 47% inhibition based on respectively 19% and 34% inhibition of the single molecules (Figure 1A). Likewise, the combination of 5.0 µM ribavirin with 0.37 µM GS-441524 resulted in 93% inhibition of HPIV-3 replication, whereas the molecules alone resulted respectively in 16% and virtually no inhibition (Figure 1D). 3D maps of the combined efficacy of ribavirin with either remdesivir or GS-441524 visualize this synergism (red areas in Figure 1C,F).

### 3.2. The Combined Treatment of Ribavirin and GS-441524 Potently Inhibits HPIV-3 Replication in Nasal Human Airway Epithelial Cell (HnAEC) Cultures

Next, the combined efficacy of ribavirin and GS-441524 against HPIV-3 in nasal human airway epithelial cell (HnAEC) cultures grown at the air-liquid interface was assessed. First, we determined which suboptimal antiviral concentrations of either molecule were to be used in the combination experiments; these were, respectively, 30 µM and 0.30 µM for ribavirin and GS-441524 (Figure 2A,B), which either alone or in combination did not result in toxic effects in the HnAEC cultures (as determined by quantifying the release of LDH per day and at the endpoint) (Figure 2C,D). Next, the combined antiviral efficacy was assessed over an extended period of 25 days whereby the molecules were added two hours before infection to the culture medium and wherein treatment was continued until day 18 p.i. Ribavirin (100 µM) or GS-4415245 (0.3 µM) alone resulted in a slight antiviral effect during the first days of the experiment, but no inhibition of viral replication was observed between day 11 p.i. and 25 p.i., the end of the experiment, across two independent experiments. In contrast, the combined treatment of ribavirin and GS-441524 resulted in a potent reduction of HPIV-3 RNA levels (Figure 2E). In Experiment 1 (Figure 2E, left panel), HPIV-3 replication was suppressed to undetectable levels throughout the experiment in the presence of the combined treatment, resulting in a reduction of 3.5–4.5 log_10_ compared to the control group. No viral rebound was observed even one week after cessation of treatment. In Experiment 2 (Figure 2E, right panel), HPIV-3 RNA levels remained suppressed below the lower limit of quantification until day 16 p.i., with a gradual rebound in 4 out of 6 cultures in the presence of combined treatment. However, RNA levels remained lower than those in the single treatment groups. Infectious virus titers were quantified as well and were found to be markedly decreased in the combined treatment group compared to those in the control and single treatment groups during the treatment period (Appendix A). No cytotoxicity was observed in any of the groups, as determined by measuring the levels of released LDH (Appendix A).

We next studied the antiviral effect of the single and combined treatments when the molecules were first added to the cultures at 4 days p.i. and after which treatment was continued for 18 consecutive days. A reduction in HPIV-3 RNA levels was observed in the combination group, but it did not reach undetectable levels (Figure 3A, Experiment 1 and Experiment 2). No cytotoxicity was observed in any of the groups, as determined by measuring the levels of released LDH (Appendix A). Next, infectious virus titers were quantified and shown to decline in the combined therapy group in both experiments 1 and 2, reaching undetectable levels at certain time points, unlike the limited reduction observed in the single-treatment groups (Figure 3B). Notably, both HPIV-3 RNA levels and infectious virus titers in the combined therapy group were significantly lower than those in the single-treatment groups at day 14 p.i. and when analyzing the area under the curve over the treatment period (Figure 3C,D).

### 3.3. The Combined Treatment of Ribavirin and GS-441524 Potently Reduces HPIV-3 Replication and Virus-Induced Lung Pathology in AG129 Mice

We next determined the antiviral effect of either ribavirin or GS-441524 (both dosed orally) alone or in combination in a mouse HPIV-3 infection model that we recently developed and validated for antiviral studies, wherein the mice are intranasally inoculated with a clinical isolate of the virus [27]. Ribavirin, at doses of either 25 mg/kg or 50 mg/kg twice daily BID, could not significantly reduce infectious viral titers in the lungs of infected AG129 mice at day 3 p.i. (a time point when peak viral titers were noted in vehicle-treated mice) (Appendix A). Because of a large variation in infectious viral lung titers observed in the 50 mg/kg, but not in the 25 mg/kg dose group, the latter dose was selected. We previously demonstrated that GS-441524 at a dose of 50 mg/kg reduces infectious HPIV3 titers to undetectable levels in the lungs of all infected mice [27]. Therefore, lower doses [2.5 mg/kg, 10 mg/kg or 25 mg/kg (BID)] of GS-441524 were selected for testing in combination with 25 mg/kg of ribavirin. The combination of 25 mg/kg ribavirin with 25 mg/kg GS-441524 was found to result in the most potent antiviral activity at day 3 p.i. (Appendix A).

We next directly compared the antiviral efficacy of either ribavirin (25 mg/kg, per dose) or GS-441524 (25 mg/kg, per dose) alone with their combination. Treatment was started one day before intranasal inoculation with HPIV-3 and was continued for either 4 or 7 consecutive days. At day 3 p.i. or day 6 p.i., mice were euthanized and lung samples were collected for quantification of viral loads and assessment of lung histopathology, respectively (Figure 4A). These timepoints were chosen since infectious viral lung titers peak at day 3 p.i., whereas lung pathology develops prominently from day 5 p.i. [27]. Ribavirin monotherapy did not result in an antiviral effect; GS-441524 monotherapy reduced infectious viral lung titers by 1.2 log_10_ TCID_50_/mg lung sample (*p* = 0.02) as compared to the vehicle-treated group. The combined treatment resulted in a reduction of 1.9 log_10_ TCID_50_/mg lung sample (*p* < 0.001) in infectious virus titers, with no detectable infectious virus in 4 out of 11 mice (Figure 4B). No significant weight loss or signs of adverse effects were observed in any group at day 6 p.i. (Appendix A). The lungs of uninfected mice were histologically normal, whereas the vehicle-treated mice presented with intra-alveolar hemorrhage, significant peri-vascular and peri-bronchial inflammation and bronchopneumonia foci. Most of the mice from both the ribavirin and GS-441524 monotherapy groups presented with limited but significant peri-vascular and peri-bronchial inflammation and intra-alveolar hemorrhage. In contrast, only mild peri-vascular inflammation in 6 out of 9 mice and mild intra-alveolar hemorrhage in 2 out of 9 mice were observed in the combined treatment group (Figure 4C). The median lung histopathology score was 5.5 in the vehicle-treated group versus 1.5 in the combined treatment group (Figure 4D); the median lung score in the uninfected untreated mice was 1.0. Although the median scores between the single-treatment groups and the combined treatment group were not vastly different, 5 out of 11 mice in the ribavirin-treated group and 3 out of 11 mice in the GS-441524-treated group had higher scores than those in the combined treatment group. Importantly, four mice in the single-treatment groups had severely affected lungs, whereas this was not the case in any of the mice that received the combined treatment. Taken together, the combined treatment of ribavirin and GS-441524 markedly reduces infectious viral titers in the lungs and protects AG129 mice against severe lung pathology.

## 4. Discussion

HPIV-3 is an important causative agent of lower respiratory tract illnesses in vulnerable populations, particularly children under 5 years, the elderly and immunocompromised individuals. There are no vaccines nor antiviral drugs available for preventing and treating HPIV-3 or other HPIVs infections. We here demonstrate that ribavirin combined with remdesivir or its parent nucleoside, GS-441524, results in potent antiviral activity against HPIV-3 infection both in vitro and in vivo.

In LLC-MK2 cells, the combined treatment of ribavirin with either remdesivir or GS-441524 resulted in a marked antiviral synergism against HPIV-3 over a relatively broad concentration range. In human nasal airway epithelial cell cultures grown at the air–liquid interface, the combined treatment of ribavirin and GS-441524, each at suboptimal or even inactive concentrations, resulted in pronounced inhibition of HPIV-3 replication. In AG129 mice, the combined treatment of ribavirin and GS-441524 resulted in a potent antiviral effect against HPIV-3, with lower or undetectable levels of infectious viruses in the lungs of more animals as compared with the single treatment groups. All of the mice in the combination group had virtually normal lung histology.

Ribavirin and remdesivir inhibit viral RNA synthesis through different mechanisms. GS-441524, the parent nucleoside of remdesivir, is intracellularly converted to its 5′-triphosphate metabolite (which can be considered a mimic of the natural substrate ATP) and inhibits viral replication by incorporation into the viral RNA by the RNA-dependent RNA polymerase followed by chain termination [35,36]. Ribavirin, once converted to its 5′-monophosphate, acts as a potent competitive inhibitor of inosine monophosphate dehydrogenase resulting in depletion of intracellular GTP pools [37]. Depletion of GTP pools is a predominant mechanism of ribavirin’s antiviral activity against many viruses, including paramyxoviruses [11,38,39]. Depleted intracellular GTP pools may in turn potentially lead to lower ATP pools, since cells can consume ATP in an effort to maintain GPP pools [40,41]. Furthermore, the 5′-triphosphate of ribavirin can result in mutation-induced viral error catastrophe and chain termination [42]. The simultaneous combined (i) inhibition of viral RNA replication by remdesivir (GS-441524), (ii) depletion of intracellular GTP pools by ribavirin, (iii) depletion of ATP pools and thus reduced competition at the level of the viral polymerase for the 5′-triphosphate of GS-441524, and (iv) error catastrophe induction by ribavirin may result in a more efficient (read synergistic) decline in viral replication than what may be expected from a solely additive antiviral effect of both molecules.

In conclusion, we demonstrate that ribavirin combined with remdesivir or GS-441524 results in a markedly more pronounced antiviral effect than what would be expected from an additive antiviral effect. This finding provides preclinical evidence to explore in clinical studies the efficacy of co-administration of ribavirin with either remdesivir or obeldesivir (which can be seen as the oral form of remdesivir) to combat HPIV infections. Likewise, such combinations may also be explored against infections with other viruses that are also susceptible to these two drugs in vitro.

## Figures and Tables

**Figure 1 viruses-17-00172-f001:**
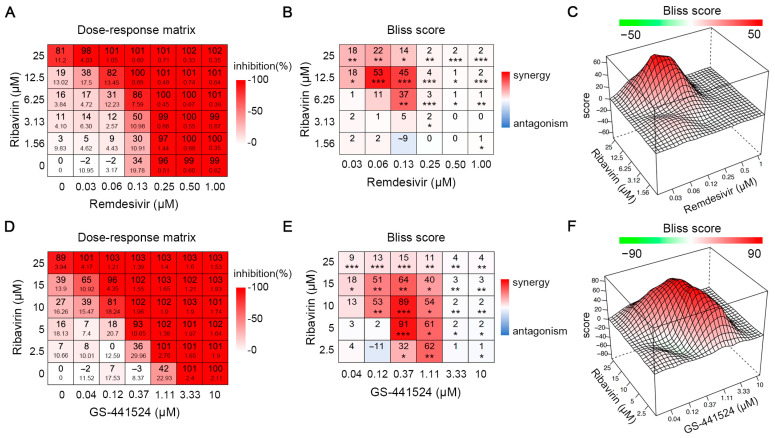
The combined antiviral effect of ribavirin with either remdesivir or GS-441524 results in synergistic antiviral efficacy against HPIV-3 replication in LLC-MK2 cells. (**A**,**D**) The matrices of inhibition of HPIV-3 replication in the presence of combined treatments. Data are from five independent experiments. (**B**,**E**) The interaction between two molecules was determined by using the Bliss independence model. Data are presented as mean values of BLISS scores. Data are analyzed using the one-sample Student’s *t*-test. *p* < 0.05, *; *p* < 0.01, **; *p* < 0.001, ***. (**C**,**F**) 3D synergy maps.

**Figure 2 viruses-17-00172-f002:**
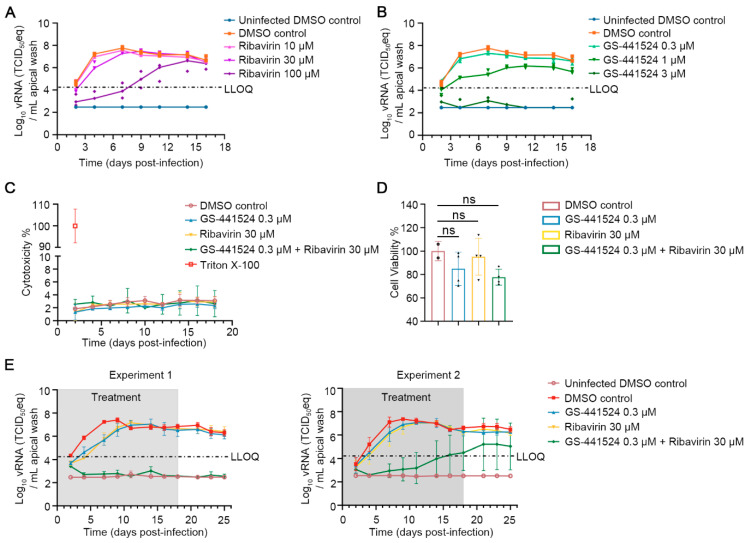
The combination of ribavirin and GS-441524 potently inhibits HPIV-3 replication in nasal human airway epithelial (HnAEC) cultures in a prophylactic setting. (**A**,**B**) The antiviral effect of ribavirin and GS-441524 at various concentrations against HPIV-3 replication. Molecules were added to the basal medium of HnAEC cultures 2 h prior to infection, with treatments continuing for 16 consecutive days. Viral RNA levels in the apical washes were quantified by RT-qPCR. Individual data from two inserts of uninfected DMSO control and three inserts of each of the other groups are presented. (**C**,**D**) The toxicity of the molecules was determined by LDH assay per day (**C**) or at the endpoint (**D**). The maximum LDH activity was determined by lysing two fresh HnAE culture inserts using Triton X-100 for 24 h. Data from four inserts are presented as mean ± SD and analyzed using one-way ANOVA with Dunnett’s multiple comparisons test. ns, nonsignificant. (**E**) Kinetics of HPIV-3 replication in the presence of single treatments with each of GS-441524, ribavirin or their combination; two independent experiments (Experiment 1 and Experiment 2). Molecules were added for 18 consecutive days, followed by further culturing with basolateral medium that did not contain inhibitors. Levels of viral RNA in apical washes were quantified by RT-qPCR. Data from three inserts of uninfected DMSO control and four inserts from each of the other groups, and data from six inserts of the combination group and four inserts from each of the other groups are presented as mean ± SD in Experiment 1 and Experiment 2, respectively. The grey boxes indicate the treatment window. LLOQ presents the lower limit of quantification. “eq” stands for equivalent, as vRNA was measured but TCID_50_-equivalent was reported.

**Figure 3 viruses-17-00172-f003:**
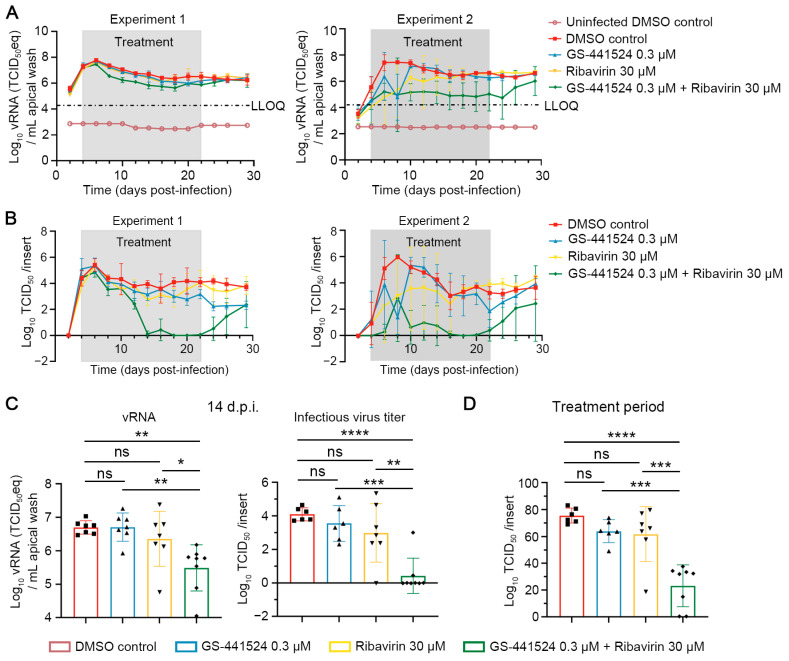
The combination of ribavirin and GS-441524 potently inhibits HPIV-3 replication in HnAEC cultures in a therapeutic setting. At day 4 p.i., the HnAEC cultures were treated with GS-441524, ribavirin or their combination. Treatment continued until day 22 p.i., after which the basal medium without molecules was refreshed. (**A**) HPIV-3 RNA levels in apical washes were quantified by RT-qPCR. LLOQ presents the lower limit of quantification. (**B**) Infectious virus titers in apical washes were determined by endpoint virus titration assay. The lower limit of detection is −0.02. Experiment 1 and 2 are two independent experiments. Data from 3–4 inserts per group are presented as mean ± SD in Experiment 1 and 2. (**C**) HPIV-3 RNA levels and infectious virus titers at 14 days post-infection (14 d.p.i.). (**D**) Infectious virus titers of area under the curve over the treatment period in Experiment 1 and 2. Data from seven or eight inserts per group across two independent experiments are presented as mean ± SD. Data were analyzed using one-way ANOVA with Tukey’s multiple comparisons test. ns, nonsignificant; *p* < 0.05, *; *p* < 0.01, **; *p* < 0.001, ***; *p* < 0.0001, ****. The grey boxes indicate treatment windows.

**Figure 4 viruses-17-00172-f004:**
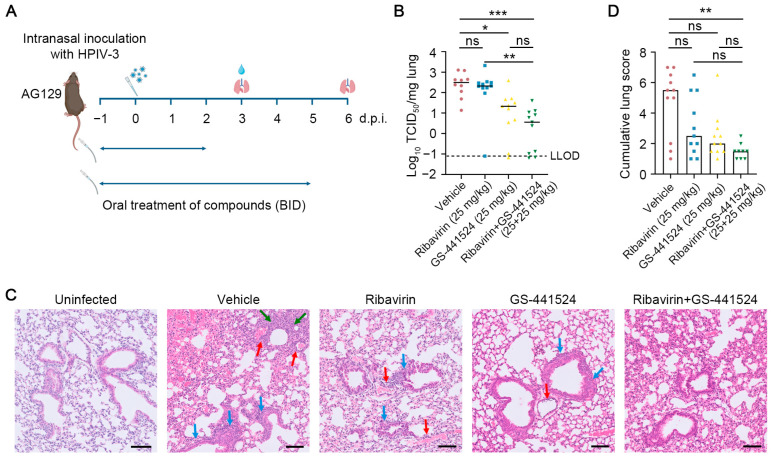
The combination of ribavirin and GS-441524 potently reduces viral loads and lung pathology upon intranasal inoculation of HPIV-3 in AG129 mice (**A**) Setup of the study. AG129 mice were orally dosed with either vehicle, ribavirin (25 mg/kg, per dose), GS-441524 (25 mg/kg, per dose) or their combination twice daily from day 1 p.i. to day 2 p.i. or day 5 p.i. At day 0, the mice were intranasally inoculated with 1.5 × 10^6^ TCID_50_ of HPIV-3. Lung samples were collected at day 3 p.i. to measure viral loads and at day 6 p.i. to assess lung histopathology. (**B**) Infectious virus titers are expressed as log_10_ TCID_50_ per milligram of lung sample. LLOD presents the lower limit of detection. Data are from two independent experiments with 10 mice in the vehicle-treated group and 11 mice in drug-treated groups. (**C**) Representative H&E-stained images revealed normal lung parenchyma in uninfected mice, intra-alveolar hemorrhage (left upper corner), significant peri-vascular (red arrows) and peri-bronchial (blue arrows) inflammation, and focus of bronchopneumonia (green arrows) in the vehicle group, focal but significant peri-vascular (red arrows) and peri-bronchial (blue arrows) inflammation in the ribavirin-treated group, very limited peri-vascular (red arrow) and peri-bronchial (blue arrows) inflammation in the GS-441524 group, and very limited and slight peri-vascular inflammation in the combination group. The samples of uninfected mice and infected mice were from the same experiment, but the staining procedure was performed at different moments. The scale bar is 100 µm. (**D**) Cumulative severity scores of lungs of all infected mice. Data are from two independent experiments with 9 mice in the combination-treated group and 11 mice in all other groups. Individual data and median values (indicated by bars) are presented in all graphs. Data were analyzed with the Kruskal–Wallis test. ns, nonsignificant; *p* < 0.05, *; *p* < 0.01, **; *p* < 0.001, ***. (**A**) was designed with BioRender.

## Data Availability

Data is contained within the article or Appendix A.

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
