# Peer review of "The Combination of GS-441524 (Remdesivir) and Ribavirin Results in a Potent Antiviral Effect Against Human Parainfluenza Virus 3 Infection in Human Airway Epithelial Cell Cultures and in a Mouse Infection Model"

_viruses, 2025, doi:10.3390/v17020172_

Round 1
Reviewer 1 Report
Comments and Suggestions for Authors
The manuscript “The Combination of GS-441524 (Remdesivir) and Ribavirin Results in a Potent Antiviral Effect Against Human Parainfluenza Virus 3 Infection in Human Airway Epithelial Cell Cultures and in a Mouse Infection Model” presents an in vivo study on the combination of two nucleoside antivirals for the treatment of Human Parainfluenza Virus Type 3 (HPIV-3) infection. The manuscript is well-written, well-organized, and effectively illustrated. The study design appears robust, and the methods are adequately described. I find the work highly interesting and suitable for publication after addressing the following minor comments:
1. In the mouse experiment, both drugs were administered at the same dose of 25 mg/kg, despite GS-441524 demonstrating a 10-fold higher activity than ribavirin in vitro. How do the selected doses correlate with the toxicity profiles of the drugs? Additional information on the effective and lethal doses (ED50 and LD50) of the drugs would provide better context for the dosing regimen used in the study.
2. Regarding the in vitro cytotoxicity data, are the cytotoxic effects of GS-441524 and ribavirin in the standard MTT assay synergistic or additive? Clarifying this would provide a more comprehensive understanding of the safety profile of the drug combination.
Author Response
The manuscript “The Combination of GS-441524 (Remdesivir) and Ribavirin Results in a Potent Antiviral Effect Against Human Parainfluenza Virus 3 Infection in Human Airway Epithelial Cell Cultures and in a Mouse Infection Model” presents an in vivo study on the combination of two nucleoside antivirals for the treatment of Human Parainfluenza Virus Type 3 (HPIV-3) infection. The manuscript is well-written, well-organized, and effectively illustrated. The study design appears robust, and the methods are adequately described. I find the work highly interesting and suitable for publication after addressing the following minor comments:
- In the mouse experiment, both drugs were administered at the same dose of 25 mg/kg, despite GS-441524 demonstrating a 10-fold higher activity than ribavirin in vitro. How do the selected doses correlate with the toxicity profiles of the drugs? Additional information on the effective and lethal doses (ED50 and LD50) of the drugs would provide better context for the dosing regimen used in the study.
We determined experimentally what the (near to highest) dose was of each of the two molecules that is not or only suboptimal active. We found that ribavirin at doses of either 25 mg/kg or 50 mg/kg twice daily (BID) does not significantly reduce viral loads in the lungs. No apparent toxicity was observed at either dose (the animals behaved normally and did not lose weight as compared to the vehicle-treated mice). However, a rather large variation in infectious viral lung titers was observed in the 50 mg/kg, but not in the 25 mg/kg dose group. These data were already included in the supplementary figure (Fig. S2A) of the original submission (now referenced in lines 260-263 of the revised manuscript). For these reasons, the lower dose, 25 mg/kg, was selected.
We previously determined (Lin et al., Nature Communications 2024) that GS-441524 at a dose of 50 mg/kg (that is very well tolerated) completely blocks viral replication, reducing infectious HPIV3 titers to undetectable levels in the lungs of all treated infected mice. Thus a dose of 50 mg/kg is not suitable for combination studies. We therefore explored the use of lower doses [2.5 mg/kg, 10 mg/kg, or 25 mg/kg (BID)] of GS-441524 for the combination studies. We observed that the combination of 25 mg/kg ribavirin with 25 mg/kg GS-441524, whereby the latter drug was at this dose suboptimal active, resulted in the most potent antiviral activity at day 3 p.i. The result was also already included as supplementary data (Fig. S2B) in the original submission (now referenced in lines 264-268 of the revised manuscript).
- Regarding the in vitro cytotoxicity data, are the cytotoxic effects of GS-441524 and ribavirin in the standard MTT assay synergistic or additive? Clarifying this would provide a more comprehensive understanding of the safety profile of the drug combination.
Potential toxic effects of ribavirin in combination with either remdesivir or GS-441524 were tested using standard MTS assays. An additive effect was observed. We refer to the table below for details. We have included a description of this finding in the Results section (lines 207-210 in the revised manuscript), accompanied by a note of “data not shown”. We leave it as an editorial decision to either include or not include these data in the manuscript, but we feel that it does not add much to the storyline.
|
Ribavirin/ µM |
25 |
30.9 |
32.1 |
32.9 |
33.4 |
32.5 |
33.0 |
35.4 |
|
12.5 |
22.7 |
23.1 |
23.6 |
24.1 |
24.2 |
26.4 |
28.2 |
|
|
6.25 |
11.2 |
12.6 |
13.1 |
13.8 |
13.4 |
11.3 |
7.6 |
|
|
3.13 |
7.3 |
7.6 |
10.2 |
10.4 |
9.5 |
5.4 |
0.0 |
|
|
1.56 |
-0.3 |
6.9 |
6.5 |
6.2 |
5.0 |
0.3 |
-4.6 |
|
|
0 |
0.0 |
3.5 |
2.7 |
3.8 |
1.4 |
-0.9 |
-8.8 |
|
|
Toxicity% |
0 |
0.03 |
0.06 |
0.13 |
0.25 |
0.5 |
1 |
|
|
Remdesivir/µM |
||||||||
|
Ribavirin/ µM |
25 |
24.3 |
26.1 |
26.9 |
28.4 |
25.9 |
24.1 |
24.3 |
|
15 |
22.3 |
23.9 |
23.5 |
24.0 |
25.0 |
21.3 |
20.4 |
|
|
10 |
18.8 |
19.3 |
20.6 |
18.5 |
18.7 |
17.9 |
15.7 |
|
|
5 |
10.4 |
8.2 |
16.3 |
15.4 |
12.8 |
11.1 |
10.6 |
|
|
2.5 |
4.0 |
8.1 |
8.7 |
11.8 |
11.6 |
9.6 |
4.7 |
|
|
0 |
0.0 |
2.3 |
3.8 |
4.3 |
5.0 |
2.8 |
1.0 |
|
|
Toxicity% |
0.00 |
0.04 |
0.12 |
0.37 |
1.11 |
3.33 |
10 |
|
|
GS-441524/µM |
||||||||
Table: Matrices of toxicity of the combined treatments (whereby ribavirin is combined with either remdesivir or GS441524). Data are mean values of three independent experiments.
Reviewer 2 Report
Comments and Suggestions for Authors
The present article is a good entry in the field of antiviral therapy. The Authors have clearly presented and investigated the activity of two known antiviral drugs, namely ribavirin and remdesivir, against HPIV type 3 . They make a comparative analysis of antiviral activity between the two drugs as monotherapy and combination therapy. The combination approach based on ribavirin and the prodrug of remdesivir led to better results over single treatment, more effectively decreasing infectious viral lung titers. The paper is well organised, references are pertinent and updated. In vitro and in vivo tests look solid and significant.
I only suggest including in the introduction a brief description of antiviral agents, besides ribavirin and remdesivir, which were investigated/repurposed against HPIV type 3. Please see the following papers as an example:
Smielewska A, Emmott E, Goodfellow I, Jalal H. In vitro sensitivity of human parainfluenza 3 clinical isolates to ribavirin, favipiravir and zanamivir. J Clin Virol. 2018 May;102:19-26. doi: 10.1016/j.jcv.2018.02.009.
Bailly B, Dirr L, El-Deeb IM, Altmeyer R, Guillon P, von Itzstein M. A dual drug regimen synergistically blocks human parainfluenza virus infection. Sci Rep. 2016 Apr 7;6:24138. doi: 10.1038/srep24138. PMID: 27053240; PMCID: PMC4823791.
Author Response
The present article is a good entry in the field of antiviral therapy. The Authors have clearly presented and investigated the activity of two known antiviral drugs, namely ribavirin and remdesivir, against HPIV type 3 . They make a comparative analysis of antiviral activity between the two drugs as monotherapy and combination therapy. The combination approach based on ribavirin and the prodrug of remdesivir led to better results over single treatment, more effectively decreasing infectious viral lung titers. The paper is well organized, references are pertinent and updated. In vitro and in vivo tests look solid and significant. I only suggest including in the introduction a brief description of antiviral agents, besides ribavirin and remdesivir, which were investigated/repurposed against HPIV type 3. Please see the following papers as an example:
Smielewska A, Emmott E, Goodfellow I, Jalal H. In vitro sensitivity of human parainfluenza 3 clinical isolates to ribavirin, favipiravir and zanamivir. J Clin Virol. 2018 May;102:19-26. doi: 10.1016/j.jcv.2018.02.009.
Bailly B, Dirr L, El-Deeb IM, Altmeyer R, Guillon P, von Itzstein M. A dual drug regimen synergistically blocks human parainfluenza virus infection. Sci Rep. 2016 Apr 7;6:24138. doi: 10.1038/srep24138. PMID: 27053240; PMCID: PMC4823791.
Thank you for this suggestion. We have included a brief description of antiviral agents besides ribavirin and remdesivir in the Introduction section (lines 47-51 in the revised manuscript). This new piece of text reads as follows: “Several antiviral drugs have been investigated for repurposing as treatments for HPIV-3 infection, including the RNA-dependent RNA polymerase (RdRp) inhibitors remdesivir, ribavirin, and favipiravir, as well as the neuraminidase inhibitor zanamivir (Smielewska et al., 2018; Bailly et al., 2016). The latter drug results in a synergistic in vitro antiviral effect when combined with the entry inhibitor suramin (Bailly et al., 2016).”